# The Long-Term Mortality Effects Associated with Exposure to Particles and NO_x_ in the Malmö Diet and Cancer Cohort

**DOI:** 10.3390/toxics11110913

**Published:** 2023-11-07

**Authors:** Henrik Olstrup, Erin Flanagan, Jan-Olov Persson, Ralf Rittner, Hanne Krage Carlsen, Leo Stockfelt, Yiyi Xu, Lars Rylander, Susanna Gustafsson, Mårten Spanne, Daniel Oudin Åström, Gunnar Engström, Anna Oudin

**Affiliations:** 1Division of Occupational and Environmental Medicine, Department of Laboratory Medicine, Lund University, 223 63 Lund, Sweden; erin.flanagan@med.lu.se (E.F.); danielastrom@hotmail.com (D.O.Å.);; 2Sustainable Health, Department of Public Health and Clinical Medicine, Umeå University, 901 87 Umeå, Sweden; 3Department of Mathematics, Stockholm University, 106 91 Stockholm, Sweden; joper@math.su.se; 4School of Public Health and Community Medicine, Institute of Medicine, Center of Registers, Sahlgrenska Academy, University of Gothenburg, 413 45 Gothenburg, Sweden; 5Occupational and Environmental Medicine, School of Public Health and Community Medicine, Institute of Medicine, Sahlgrenska Academy, University of Gothenburg, 413 90 Gothenburg, Sweden; 6Department of Occupational and Environmental Medicine, Sahlgrenska University Hospital, 413 90 Gothenburg, Sweden; 7Environment Department, City of Malmö, 205 80 Malmö, Sweden; 8Department of Clinical Sciences at Malmö, CRC, Lund University, 221 00 Lund, Sweden

**Keywords:** air pollution, long-term exposure, particles, nitrogen oxides, Cox regression, proportional hazard, hazard ratio

## Abstract

In this study, the long-term mortality effects associated with exposure to PM_10_ (particles with an aerodynamic diameter smaller than or equal to 10 µm), PM_2.5_ (particles with an aerodynamic diameter smaller than or equal to 2.5 µm), BC (black carbon), and NO_x_ (nitrogen oxides) were analyzed in a cohort in southern Sweden during the period from 1991 to 2016. Participants (those residing in Malmö, Sweden, born between 1923 and 1950) were randomly recruited from 1991 to 1996. At enrollment, 30,438 participants underwent a health screening, which consisted of questionnaires about lifestyle and diet, a clinical examination, and blood sampling. Mortality data were retrieved from the Swedish National Cause of Death Register. The modeled concentrations of PM_10_, PM_2.5_, BC, and NO_x_ at the cohort participants’ home addresses were used to assess air pollution exposure. Cox proportional hazard models were used to estimate the associations between long-term exposure to PM_10_, PM_2.5_, BC, and NO_x_ and the time until death among the participants during the period from 1991 to 2016. The hazard ratios (HRs) associated with an interquartile range (IQR) increase in each air pollutant were calculated based on the exposure lag windows of the same year (lag0), 1–5 years (lag1–5), and 6–10 years (lag6–10). Three models were used with varying adjustments for possible confounders including both single-pollutant estimates and two-pollutant estimates. With adjustments for all covariates, the HRs for PM_10_, PM_2.5_, BC, and NO_x_ in the single-pollutant models at lag1–5 were 1.06 (95% CI: 1.02–1.11), 1.01 (95% CI: 0.95–1.08), 1.07 (95% CI: 1.04–1.11), and 1.11 (95% CI: 1.07–1.16) per IQR increase, respectively. The HRs, in most cases, decreased with the inclusion of a larger number of covariates in the models. The most robust associations were shown for NO_x_, with statistically significant positive HRs in all the models. An overall conclusion is that road traffic-related pollutants had a significant association with mortality in the cohort.

## 1. Introduction

Exposure to air pollution has, in a wide range of studies, been shown to have detrimental long-term health effects. According to the World Health Organization, ambient (outdoor) air pollution is estimated to have caused around 4.2 million premature deaths worldwide in 2019 [1]. The long-term mortality effects associated with exposure to PM_2.5_ (particles with an aerodynamic diameter smaller than or equal to 2.5 µm) and PM_10_ (particles with an aerodynamic diameter smaller than or equal to 10 µm) were analyzed in several previous studies. In a meta-analysis, reporting pooled estimates of the results from different studies around the world, natural-cause mortality was associated with long-term exposure to both PM_2.5_ and PM_10_ [2]. Increases in all-cause mortality associated with exposure to PM_2.5_ were found in other meta-analyses, including studies from a wide range of geographic areas [3,4]. In Europe, associations between mortality and long-term exposure to PM_2.5_ were found in two multi-cohort studies [5,6]. Furthermore, in a systematic review based on cohort studies conducted in Australia, Mainland China, Hong Kong, Taiwan, and South Korea, an association between long-term exposure to PM_2.5_ and mortality was found [7]. Considering Sweden specifically, a multi-cohort study demonstrated a statistically significant increase in natural mortality associated with long-term exposure to PM_10_, and for PM_2.5_, the effect was positive but not statistically significant [8].

The long-term effects on mortality associated with exposure to BC (black carbon), colloquially called soot particles, were analyzed in a few studies. BC refers to the carbonaceous fraction of particles originating from incomplete combustion, and they are usually measured by light absorption. In a large population-based French cohort, with data collected from 1989 to 2017, long-term exposure to BC was associated with all-cause mortality [9]. Long-term exposure to black smoke (BS), a previously common technique to measure soot particles, was used in older cohort studies. One such cohort study, spanning seven cities in France, found an association between non-accidental mortality and exposure to BS (assessed in the 1970s) over a 25-year study period [10]. Similarly, a Dutch cohort study on long-term exposure to traffic-related air pollutants, analyzed from 1987 to 1996, demonstrated an association between natural mortality and exposure to BS [11].

In addition to particulate air pollution, the mortality effects associated with long-term exposure to gaseous pollutants, including NO_x_ (nitrogen oxides), the sum of NO (nitrogen monoxide) and NO_2_ (nitrogen dioxide), and NO_2_ itself, were investigated in previous studies. For instance, associations between mortality and long-term exposure to NO_2_ were shown in several meta-analyses based on pooled estimates from many original studies [3,12,13,14,15]. The mortality effects associated with long-term exposure to NO_x_ were analyzed in a cohort in Gothenburg, Sweden, during the period from 1973 to 2007, and associations were found for three different time lag windows [16].

The long-term health effects associated with exposure to air pollutants were explored in previous studies in a cohort in Malmö, Southern Sweden, called “The Malmö Diet and Cancer Cohort (MDC)”. In this cohort, air pollution concentrations were assessed at the participants’ home addresses. The associations between cardiovascular diseases and exposure to NO_x_ and particles were analyzed in the MDC cohort during the period from 1991 to 2016. Several statistically significant hazard ratios between exposure to NO_x_ or particles and cardiovascular outcomes were found [17]. The long-term associations between a number of biomarkers and exposure to NO_x_ and particles were also analyzed in the MDC cohort, where several statistically significant associations were found [18]. Statistically significant associations between long-term exposure to NO_x_ and particles and chronic kidney disease were also shown in the MDC cohort [19].

In the present study, the aim was to analyze the associations between long-term exposure to PM_10_, PM_2.5_, BC, and NO_x_ and mortality in the MDC cohort.

## 2. Materials and Methods

### 2.1. Description of the Study Population and the Study Period

This study is based on “The Malmö Diet and Cancer Cohort”, abbreviated to MDC, which included those residing in Malmö, Sweden, born during the period from 1923 to 1950. Initially, the main purpose of this cohort was to clarify whether a Western diet is associated with specific types of cancer while controlling for other relevant lifestyle factors [20]. The MDC cohort was described in more detail in a number of previous studies [20,21,22,23].

At enrollment, during the period from 1991 to 1996, the participants underwent health screening in terms of questionnaires about lifestyle and diet, clinical examination, and blood sampling where a total number of 30,438 participants were selected. The cohort participants were followed from enrollment until death, or until the end of year 2016 for those who survived the entire study period. Among the total number of 30,438 participants, 17,551 (57.5%) survived the entire study period, 12,663 (41.6%) were deceased during the study period, and 264 (0.9%) moved out of the study area. The cohort participants who survived the entire study period and those who moved out of the study area were censored.

### 2.2. Air Pollution Exposure Assessment

The modeled concentrations of PM_10_, PM_2.5_, BC, and NO_x_ at the cohort participants’ home addresses were used to calculate exposure. The air pollution concentrations in Malmö (18 km × 18 km) were modeled by the Environmental Department of the City using EnviMan software package (Opsis AB, Sweden) (Figure 1). Each pollutant was modeled as the sum of local emissions from traffic exhaust, non-exhaust traffic emissions (mechanically generated particles from road wear, tire wear, and brake wear, including resuspension), heating, shipping, industry, households, and long-distance transported emissions. The modeled concentrations were presented as grids with a spatial resolution of 50 m × 50 m. The participants’ addresses during the period from 1991 to 2016 were retrieved and geocoded by Statistics Sweden. The mean concentrations of PM_10_, PM_2.5_, BC, and NO_x_ for each year during the period from 1991 to 2016 were assigned to each participant’s home address. In order to capture the lag effects associated with exposure and mortality, the modeled concentrations were based on the exposure lag windows of the same year as the event (death) (lag0), 1–5 prior to the event (lag1–5), and 6–10 years prior to the event (lag6–10) (Table 1). The rationale for incorporating different lags is to assess the significance of exposure during distinct time frames.

### 2.3. Outcome Assessment

Natural-cause mortality (mortality due to causes other than injuries and trauma) was used as an outcome variable, and data were retrieved from the National Cause of Death Register. Natural-cause mortality was defined on the basis of the underlying cause of death according to the International Classification of Diseases, Tenth Revision (ICD-10: A00–R99).

### 2.4. Covariates

All covariates were collected at enrollment. Adjustments for potential confounders were made with age and gender as two basic covariates. Education level (low < 9 years, medium = 9–12 years, and high > 12 years) and cohabitation (yes or no) were included as adjustments for socio-economic factors. Smoking habits were included as never, former, or current smoker (occasional and regular). Systolic and diastolic blood pressure as well as the use of antihypertensive medications (yes or no) were included as covariates. Alcohol consumption was included as the self-reported quantity in terms of grams per day. Physical activity was categorized into low, medium, or high. Waist/hip ratio was also included as a covariate. Table 2 and Table 3 present the different covariates included in the models. Table 2 presents the continuous variables included in the models, and Table 3 presents the categorical variables included in the models.

### 2.5. Statistical Analysis

Cox proportional hazard models with time-varying air pollution exposure adjusting for possible confounders (covariates) were used to calculate the associations between long-term exposure to PM_10_, PM_2.5_, BC, and NO_x_ and time until death among the MDC cohort participants. The time since the start of the study for each study participant was used on the time axis.

Three statistical models with varying levels of adjustment were used. Model 1 represented a crude model with air pollutants adjusted only for age at enrollment and gender. Model 2 additionally adjusted for smoking habits, educational level, and cohabitation. In addition to the covariates included in Model 1 and Model 2, Model 3 also adjusted for systolic and diastolic blood pressure, self-reported alcohol consumption, physical activity, waist/hip ratio, and the use of antihypertensive medications.

Two-pollutant models were included in cases where the pairwise air pollutants were not so highly correlated (i.e., with a correlation coefficient smaller than 0.8) that multicollinearity could occur. Multicollinearity was tested by creating a multiple regression based on all variables in each model, and the variance inflation factor (VIF) was then calculated. A VIF value below three was accepted for inclusion in two-pollutant models. The VIF values of the air pollutants were in some cases above the value of three. Regarding the other covariates, the VIF values were below three in all cases and were, therefore, not considered to pose a multicollinearity problem. Table A8 in Appendix A presents correlation coefficients (Pearson) between the different air pollutants based on the exposure lag windows of the same year (lag0), 1–5 years (lag1–5), and 6–10 years (lag6–10).

All analyses were performed using STATA 17.0 (StataCorp, College Station, TX, USA).

## 3. Results

### 3.1. Hazard Ratios Associated with Exposure

Figure 2, Figure 3, Figure 4, Figure 5, Figure 6 and Figure 7 present hazard ratios (HRs) with 95% confidence intervals (CIs) for natural-cause mortality associated with exposure to PM_10_, PM_2.5_, BC, and NO_x_ for lag0, lag1–5, and lag6–10 in both single- and two-pollutant models with adjustments for all covariates (Model 3). The HRs correspond to an interquartile range (IQR) increase in each air pollutant. The complete results from the Cox regressions based on all models (Models 1–3) and all lag windows in both single- and multi-pollutant models are presented in Table A1, Table A2 and Table A3.

### 3.2. Sensitivity Analysis

In order to determine if there were any differences in the calculated HRs with respect to age, an age-stratified analysis was performed. Six different age groups at enrollment were generated: Group 1 ≤ 50 years; 50 years < Group 2 ≤ 55 years; 55 years < Group 3 ≤ 60 years; 60 years < Group 4 ≤ 65 years; 65 years < Group 5 ≤ 70 years; and Group 6 > 70 years. The Cox regression model with adjustments for all covariates (Model 3) was applied for each age group based on lag1–5 in both single- and two-pollutant models (see Table A4 and Table A5 in Appendix A). The HRs were in general largest and most robust in Groups 4 and 5.

The regression models were also divided into two different periods in terms of survival time; one period for those who passed away within ten years from when the study started, and one period for those who survived more than ten years. The Cox regression model with adjustments for all covariates (Model 3) was applied for each survival time period based on lag1–5 in both single- and two-pollutant models (see Table A6 and Table A7 in Appendix A). Similar to the main models (Table A1, Table A2 and Table A3), the HRs for PM_10_, PM_2.5_, and BC did not remain statistically significant in the two-pollutant models together with NO_x_. The HRs for NO_x_ were more stable and robust for those who survived more than ten years.

The linearity of the exposure–response relationship was tested by including a quadratic polynomial for NO_x_ in the most adjusted model (Model 3) at lag1–5. The coefficient of this quadratic polynomial ended up close to zero, and the original coefficient of NO_x_ and its *p*-value changed only slightly. This indicates that there was a reasonable linear relationship between exposure to NO_x_ and survival time in this cohort.

## 4. Discussion

### 4.1. Key Results

In this cohort study from southern Sweden with >30,000 participants from the general population and up to 25 years of follow-up, clear associations were observed between mortality and long-term residential exposure to NO_x_. NO_x_ was the only air pollutant that exhibited statistically significant associations in all models and for all lags in the main models. The associations for the other air pollutants (PM_10_, PM_2.5_, and BC) did not remain statistically significant in the two-pollutant models with NO_x_. Adjustments for lifestyle factors and other possible confounding factors had some impact on the size of the effect estimates, but, in general, the associations remained.

### 4.2. The Calculated Hazard Ratios and Possible Explanations

Among the analyzed air pollutants, the HRs associated with NO_x_ were the most clear and robust. NO_x_ is an indicator of traffic emissions, and traffic is typically the major source of NO_x_ in urban areas [24]. The HRs associated with PM_10_, PM_2.5_, and BC were, in most cases, statistically significant in the single-pollutant models. However, in the two-pollutant models with NO_x_, no statistically significant positives were shown for PM_10_, PM_2.5_, or BC. Both NO_x_ and BC originate from combustion processes. However, the correlation coefficients between NO_x_ and BC, presented in Table A8, were in the range of 0.4 and 0.5, indicating that they largely originate from different sources. Besides traffic, there are several sources that generate combustion-related emissions. BC can also originate from heating, shipping, industry, households, and long-distance transported emissions, and may therefore be modestly correlated with NO_x_, which mainly originates from traffic. In the main models (Table A1, Table A2 and Table A3), the HRs associated with BC were also statistically significant in all single-pollutant models and all two-pollutant models, except for those including NO_x_.

When comparing the HRs for NO_x_ in this study with the results for NO_2_ in the meta-analyses (similar increase in concentration) presented in the introduction section [3,12,13,14,15], the HRs in this study were, in general, larger. The HRs for PM_10_ and PM_2.5_ in this study were somewhat unclear and in many cases not statistically significant, which makes it difficult to compare these with the HRs in the meta-analyses where the coefficients for PM_2.5_ were positive and statistically significant [2,3,4,5,6,7]. Regarding soot particles (BC), comparisons between studies are difficult due to the lack of uniform measurement methods [25].

In the age-stratified analyses presented in Table A4 and Table A5, no consistent patterns were shown. However, a certain pattern in terms of clearer and more robust HRs for Groups 4 and 5 was shown in several analyses. Groups 4 and 5 represent those in the age groups 60–65 years and 65–70 years at enrollment, respectively. The HRs in the younger age groups and those in the oldest group (>70 years) were somewhat less clear and robust. Smaller effects in the younger age groups can be expected. Smaller effects in the oldest age group are somewhat unexpected. However, the oldest age group participants (>70 years at enrollment) were in some cases older than 90 years old at the end of the cohort time period, and it is possible that air pollution exposure could have a relatively smaller impact on survival time when other age-related causes of death become more apparent.

Considering the different lags, the HRs for particles (PM_10_, PM_2.5_, and BC) in the single-pollutant models were, in general, more robust and stable at lag1–5 and lag6–10 compared to lag0. However, the HRs for NO_x_ in the single-pollutant models were in the same order of magnitude for all lag windows. The larger effects at lag1–5 and lag6–10 compared to lag0 for particles indicate that there could be a noticeable delay effect between exposure and mortality. Another possible explanation is that the particles were more toxic further back in time. When comparing the HRs between two survival time periods (0–10 years and >10 years) shown in Table A6 and Table A7, the HRs for particles were, in general, larger at the survival time period of 0–10 years, while the HRs for NO_x_ were larger for the survival time period of >10 years. This could possibly mean that PM_10_ and PM_2.5_ were more toxic during the first ten years of the cohort period. However, when considering the trends in concentrations of the analyzed air pollution during the time period of this cohort, a decreasing trend was shown for NO_x_, while no obvious patterns were shown for PM_10_, PM_2.5_, and BC [19,26]. This indicates that the trends in concentrations cannot explain the differences in the HRs between different lags and survival times.

When considering all analyses in this study, the HRs for PM_10_ and PM_2.5_ were in many cases statistically significant in the single-pollutant models, but they were non-significant, or in some cases negatively statistically significant, in the two-pollutant models. Unspecified particulate matter (PM_10_ and PM_2.5_) does not constitute a uniform measure of particles with respect to their chemical composition and physical properties. They originate from a variety of sources that may have large spatial and temporal variations within a city. Given that this study has analyzed the exposure effect over a period of up to 25 years, the chemical composition and physical properties of the particles, and likewise their toxic potential, to which the cohort participants were exposed, most likely varied greatly during this time. Seasonal variations in the above-mentioned factors can also be assumed to have occurred. Road dust, which is most common during springtime, was found to be particularly harmful to human health [27]. Indeed, exposure to PM_2.5–10_ and PM_10_ were associated with increased mortality during springtime, but not during the rest of the year, as was shown in time-series studies performed in Stockholm [28,29]. The low correlations between NO_x_ and PM (both PM_10_ and PM_2.5_) in this study (Table A8) indicate that PM_10_ and PM_2.5_ did not to any great extent originate from traffic. The toxic fractions of PM, including road dust, could be more highly correlated with NO_x_ and may therefore not contribute much to the HRs when unspecified PM are used as exposure metrics.

### 4.3. NO_x_ as an Indicator for Other Harmful Exposures

The largest and most robust hazard ratios in this study were shown for NO_x_; however, some uncertainty remains regarding the toxicity of NO_x_ itself. Experimental studies with humans have demonstrated noticeable health effects after short-term exposure to NO_2_ at concentrations at or above 400 µg m^−3^, and health effects among patients with mild asthma could not be detected at concentrations below 200 µg m^−3^ [30]. Based on a review of several long-term studies on NO_2_ exposure, increased mortality was suggested above a threshold value of 20 µg m^−3^ [31]. The modeled mean concentrations of NO_x_ (NO + NO_2_) during the study period in this study were in the range of 25–30 µg m^−3^, and a large number of study participants may therefore have been exposed to concentrations of NO_2_ that exceeded 20 µg m^−3^ during a long period of time. However, the toxicity of NO_2_ itself, and its impact on mortality, was addressed in two literature reviews. One study indicated that there is an independent effect on long-term mortality associated with exposure to NO_2_ [13]. Contrary, the other study indicated that the greater the demands placed on the studies, the less support there is for an independent effect on long-term mortality associated with exposure to NO_2_ [15]. It is thus uncertain to what extent NO_x_ itself would have caused the robust and statistically significant associations with mortality observed in the present study. Indeed, some other component(s) of vehicle exhaust emissions, with similar dispersion patterns, may have been driving the negative health effects.

As previously discussed, NO_x_ is an established proxy for road traffic exhaust emissions. Vehicle exhaust emissions are not a homogeneous substance; however, they are comprised of harmful components other than NO_x_. For example, high correlations between NO_x_ and particle number count (PNC) were shown in Gothenburg, Sweden [32], and between NO_2_ and PNC in Stockholm, Sweden [33]. NO_x_ can, therefore, be considered a marker for PNC. PNC, in turn, is a marker for ultrafine particles (particles with an aerodynamic diameter smaller than or equal to 100 nm in all dimensions). Due to their small size and large surface area in relation to volume, ultrafine particles are believed to be more toxic than larger particles [34]. From a health perspective, particles smaller than 300 nm are especially important since they are capable of diffusing rapidly in the airway mucus through the mucus pores [35]. Ultrafine particles may, thus, be a contributing factor to the negative health effects of exposure to traffic exhaust emissions. However, fine and ultrafine particles from traffic are not only emitted from combustion processes but can also be derived from brake and tire wear [36].

Traffic noise is another health risk that can be correlated with exposure to NO_x_. The correlation between NO_x_ and noise in urban areas is determined by several factors. The short-term correlations between NO_x_ or NO_2_ and noise were analyzed in a study based on 103 urban sites with varying traffic, environment, and infrastructure characteristics. Factors that largely determined the degree of correlation were the number of lanes on the closest road, the number of cars and trucks during noise sampling, and the presence of major intersections [37]. Based on a systematic review and meta-analysis [38], the associations between long-term exposure to traffic noise and mortality were weak, except for mortality related to ischemic heart disease. With this, the authors suggested a possible threshold of 53 decibels for cardiovascular mortality from road traffic noise [38]. In a cohort study from Gothenburg, Sweden, positive but non-significant associations were found for cardiovascular mortality and morbidity and long-term residential exposure to noise above 60 decibels, compared to 50 decibels, after adjusting for air pollution exposure [39]. As traffic noise has not been included as a covariate in this study, it is not possible to draw conclusions regarding its impact on mortality among the MDC cohort participants.

In summary, the association between NO_x_ and mortality in this study is clear and robust. However, NO_x_ itself is not likely to be the main driver of this association. Traffic noise is also not expected to be responsible for the observed association, and there is a great need for studies that disentangle the effects of traffic-related noise and traffic-related air pollution on ill health. The ultrafine particles originating from both exhaust and from abrasion of tires, road surfaces, and brakes are probably an important factor, but more research is needed to confirm this. As road traffic is the main source of anthropogenic NO_x_ emissions in Europe [40], NO_x_ as well as its components and correlates will continue to be important.

### 4.4. Strengths and Limitations of This Study

A strength of this study is that it includes more than 30,000 participants and 25 years of follow-up, which provides a robust statistical basis both in terms of measurement data and number of participants. Exposure misclassification, a possible limitation, must be considered in all epidemiological studies on long-term exposure to air pollution. In this study, exposure was modeled at the participants’ home addresses, and other sources of air pollution, such as occupational and/or indoor exposure, were not considered. This is regarded as standard practice in epidemiological studies of air pollution and its health effects. Consequently, residential mobility among the study participants and its impact on exposure to air pollutants could not be taken into account. However, a previous study in another part of Sweden demonstrated that residential mobility does not seem to cause major exposure misclassification [41]. Also, exposure misclassification would have to depend on mortality to cause bias in the present study. Exposure misclassification (assumed to be non-differential) may have reduced the precision of the estimates, but low precision is not considered to be a plausible explanation for the results in the present study.

### 4.5. Future Research Needs and Policy Implications

The results of this study support traffic-related air pollution as an important environmental exposure with respect to premature mortality. While the number of epidemiological studies linking traffic emissions to adverse health effects continues to grow, the relative impact of specific components of pollutant mixtures generated by combustion engines has largely been overlooked. For instance, the health effects associated with exposure to specific exhaust components, e.g., particle-bound or free volatile organic compounds (VOCs) and polycyclic aromatic hydrocarbons (PAHs), remain to be clarified. The role of metals that originate from engine abrasion, lubrication oils, and the fuels themselves [42] that then bind to the exhaust particles also needs to be further explored.

Additionally, the characteristics and effects of non-exhaust emissions are becoming increasingly important. Mechanically generated particles from road abrasion and brakes are present in the coarse fraction (2.5–10 µm), the fine fraction (≤2.5 µm), and the ultrafine fraction (≤100 nm) [36]. Epidemiological studies analyzing the health effects associated with different chemical components of particles are not abundant. However, a time-series study conducted in the U.S., focusing on the relative risks for cardiovascular and respiratory hospital admissions associated with different chemical compositions of PM_2.5_, showed that elemental carbon, vanadium, and nickel contents were associated with an increased risk of hospital admissions [43]. Moreover, a literature review on exposure to road dust particles demonstrated serious health effects, especially for the respiratory system [27]. The components of road dust that were most frequently referenced in the reviewed studies were platinum, rhodium, bohrium, aluminum, zinc, vanadium, and polycyclic aromatic hydrocarbons [27]. Regarding brake wear particles, a toxicological study based on cell models demonstrated that brake abrasion particles and diesel exhaust particles are equally capable of damaging pulmonary cells [44].

With this, future epidemiological studies on the health effects of road traffic-related emissions should aim to include ultrafine particles and carefully adjust for different components using high-quality air pollution data.

From a policy point of view, reducing emissions from traffic has not always given rise to improved air quality in Europe and the U.S., despite long-established and progressively stringent tailpipe emission limitations. Thus, the health effects associated with different components of traffic-related emissions from combustion engines, including fuel qualities like aromatic content and metal content, still need to be addressed. These aspects influence the harmfulness of PM emissions not only from diesel, but also from gasoline- and ethanol-powered vehicles, and non-road machinery. Air pollution’s chemical components and physical composition are further modified by atmospheric processes, making their regulation more difficult. Future studies need to address the above-mentioned uncertainties.

## 5. Conclusions

In this cohort study, with roughly 30,000 participants from the general population and almost 25 years of follow-up, clear associations were observed between natural-cause mortality and long-term exposure to modeled concentrations of NO_x_ at the residential addresses. The robust hazard ratios for NO_x_ indicate that traffic-related air pollution had a significant association with mortality in the MDC cohort. However, it is uncertain to what extent NO_x_ exposure in itself is the main driver of these clear and robust hazard ratios, or if it is instead an indicator of combustion-related air pollutants including ultrafine particles and their toxic components, or road traffic noise. Hence, further research is needed to clarify the importance of specific exposures related to road traffic, air pollutants, and noise, especially ultrafine particles, their chemical components, and their toxic potential.

## Figures and Tables

**Figure 1 toxics-11-00913-f001:**
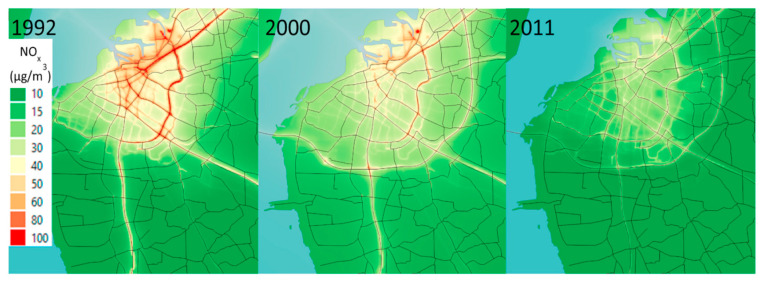
The modeled concentrations (µg m^−3^) of NO_x_ in the study area in Malmö during 1992, 2000, and 2011. Source: Carlsen et al. (2022) [17].

**Figure 2 toxics-11-00913-f002:**
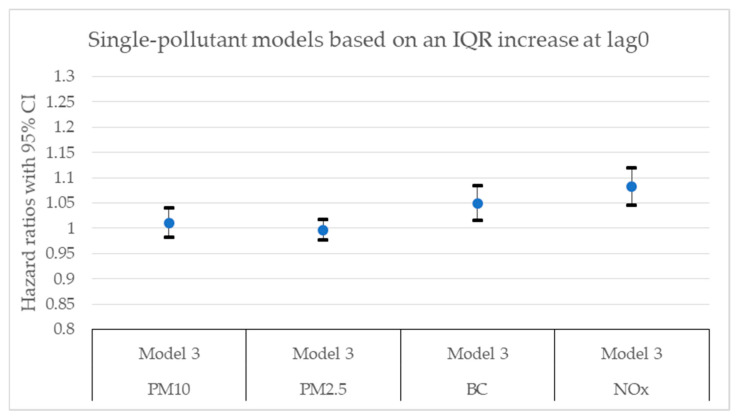
Hazard ratios with 95% confidence intervals (CI) for the associations between natural-cause mortality and exposure to PM_10_, PM_2.5_, BC, and NO_x_ based on lag0 in single-pollutant models with adjustments for all covariates (Model 3).

**Figure 3 toxics-11-00913-f003:**
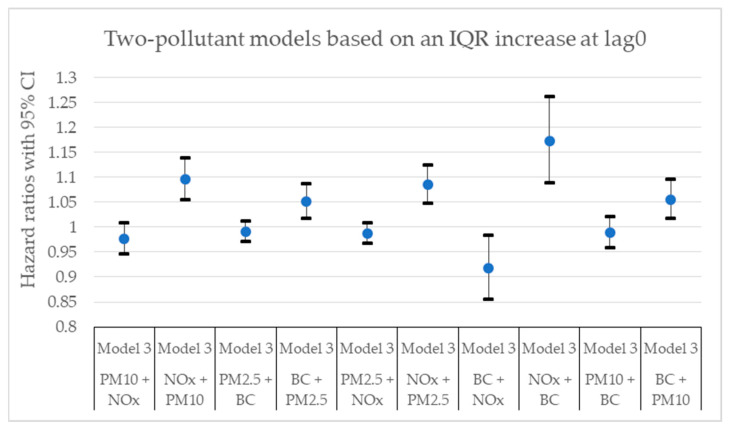
Hazard ratios with 95% confidence intervals (CI) for the associations between natural-cause mortality and exposure to PM_10_, PM_2.5_, BC, and NO_x_ based on lag0 in two-pollutant models with adjustments for all covariates (Model 3). The hazard ratios refer to the pollutant listed first.

**Figure 4 toxics-11-00913-f004:**
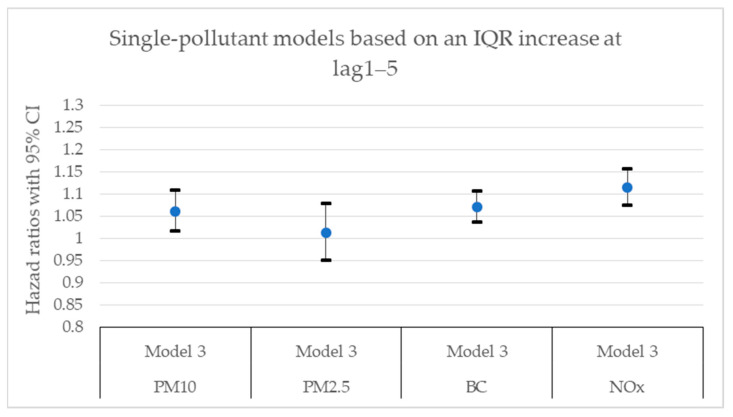
Hazard ratios with 95% confidence intervals (CI) for the associations between natural-cause mortality and exposure to PM_10_, PM_2.5_, BC, and NO_x_ based on lag1–5 in single-pollutant models with adjustments for all covariates (Model 3).

**Figure 5 toxics-11-00913-f005:**
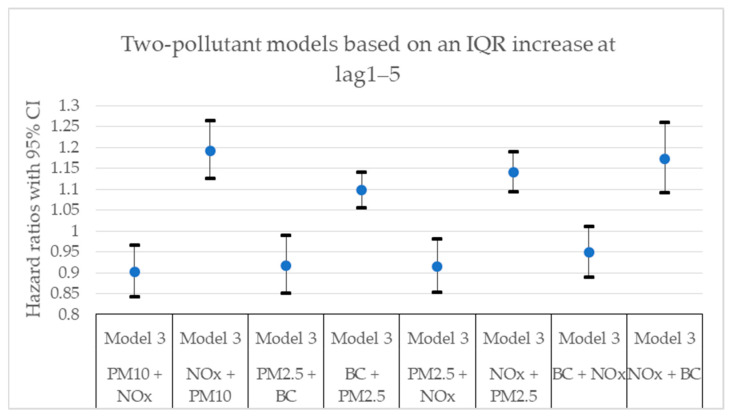
Hazard ratios with 95% confidence intervals (CI) for the associations between natural-cause mortality and exposure to PM_10_, PM_2.5_, BC, and NO_x_ based on lag1–5 in two-pollutant models with adjustments for all covariates (Model 3). The hazard ratios refer to the pollutant listed first.

**Figure 6 toxics-11-00913-f006:**
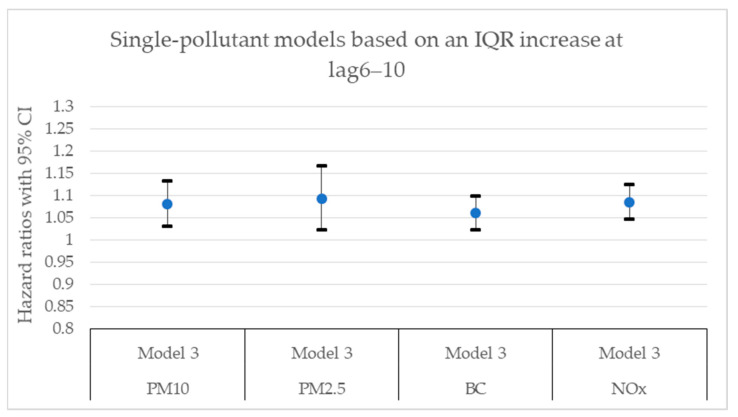
Hazard ratios with 95% confidence intervals (CI) for the associations between natural-cause mortality and exposure to PM_10_, PM_2.5_, BC, and NO_x_ based on lag6–10 in single-pollutant models with adjustments for all covariates (Model 3).

**Figure 7 toxics-11-00913-f007:**
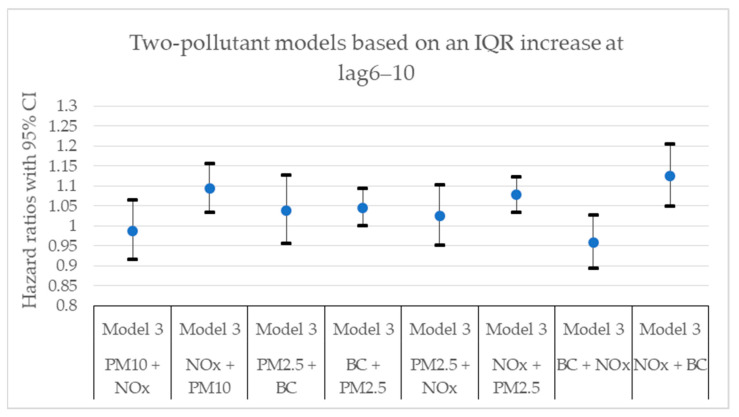
Hazard ratios with 95% confidence intervals (CI) for the associations between natural-cause mortality and exposure to PM_10_, PM_2.5_, BC, and NO_x_ based on lag6–10 in two-pollutant models with adjustments for all covariates (Model 3). The hazard ratios refer to the pollutant listed first.

**Table 1 toxics-11-00913-t001:** The modeled concentrations (µg m^−3^) of air pollutants in the study area during the period from 1991 to 2016.

Air Pollutant	Number of Obs.	Mean	SD	Median	25 Percentile	75 Percentile	Min.	Max.
PM_10_ lag0	552,608	15.9	2.2	15.7	14.5	17.2	9.7	27.6
PM_2.5_ lag0	552,608	10.9	1.8	10.8	9.8	11.4	6.6	18.4
BC lag0	552,608	1.0	0.1	1.0	0.9	1.1	0.7	1.9
NO_x_ lag0	552,608	26.5	8.8	24.9	20.8	30.5	6.8	130.0
**Air pollutant**	**Number of obs.**	**Mean**	**SD**	**Median**	**25** **percentile**	**75** **percentile**	**Min.**	**Max.**
PM_10_ lag1–5	534,059	15.8	1.5	15.9	14.8	16.9	11.2	25.8
PM_2.5_ lag1–5	534,059	10.9	0.9	10.8	10.1	11.8	8.8	14.9
BC lag1–5	534,059	0.9	0.1	1.0	0.9	1.0	0.7	1.7
NO_x_ lag1–5	534,059	28.3	10.0	26.3	21.7	33.0	7.8	134.0
**Air pollutant**	**Number of obs.**	**Mean**	**SD**	**Median**	**25** **percentile**	**75** **percentile**	**Min.**	**Max.**
PM_10_ lag6–10	417,278	15.6	1.6	15.6	14.5	16.8	11.2	24.6
PM_2.5_ lag6–10	417,278	10.7	0.9	10.6	10.0	11.5	8.8	13.8
BC lag6–10	417,278	0.9	0.1	0.9	0.9	1.0	0.7	1.7
NO_x_ lag6–10	417,278	30.1	10.6	28.1	22.7	35.4	8.6	134.0

**Table 2 toxics-11-00913-t002:** The continuous covariates included in the calculations.

Covariate	Number of Obs.	Mean	SD	Min.	Max.
Age at enrollment	30,438	58.0	7.6	44.5	73.6
Systolic blood pressure	30,389	141.1	20.1	61	240
Diastolic blood pressure	30,386	85.6	10.0	40	150
Waist/Hip ratio	30,362	0.85	0.1	0.4	1.9
Alcohol consumption (g day^−1^)	28,228	10.7	12.7	0	194

**Table 3 toxics-11-00913-t003:** The categorical covariates included in the calculations.

Covariate	Number of Obs.	Category 1	Category 2	Category 3
Gender	30,438	Male (39.8%)	Female (60.2%)	-
Smoking status	28,557	Never smoker (37.9%)	Former smoker (33.8%)	Current smoker (28.3%)
Educational level	28,492	Elementary school (42.0%)	High school (35.0%)	College (23.0%)
Cohabitation	28,554	Yes (75.4%)	No (24.6%)	-
Physical activity	30,164	Low (33.1%)	Medium (33.3%)	High (33.5%)
Antihypertensive drugs	28,446	Yes (17.8%)	No (82.2%)	-

## Data Availability

The data that support the findings of this study are available from the Lund University Medical Faculty—Malmo Diet and Cancer Cohort, but restrictions apply to the availability of these data, which were used under license for the current study, and are not publicly available. Data are however available from the authors upon reasonable request and with permission of Lund University Medical Faculty—Malmo Diet and Cancer Cohort. Ethical approval from The Swedish Ethical Review Authority is needed to access the data.

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
