# Peer review of "The Long-Term Mortality Effects Associated with Exposure to Particles and NOx in the Malmö Diet and Cancer Cohort"

_toxics, 2023, doi:10.3390/toxics11110913_

Round 1

Reviewer 1 Report

Comments and Suggestions for Authors

toxics-2597533

Long-Term Mortality Effects Associated with Exposure to Particles and NOx in the Malmö Diet and Cancer Cohort

This scientific study investigates the long-term mortality effects associated with exposure to various air pollutants, including PM10, PM2.5, black carbon (BC), and nitrogen oxides (NOx), in a cohort of individuals in southern Sweden over a 25-year period (1991–2016). The research employs rigorous statistical analyses and presents several key findings and implications for future research and policy.

The abstract provides a clear overview of the study's objectives, methods, and key results. The introduction sets the stage effectively by introducing the research question and the context of the study. However, it could be improved further by providing more background on the health and environmental impacts of the studied air pollutants. The methods section provides a comprehensive description of data collection, exposure assessment, and statistical analysis.

The discussion section effectively interprets the results and offers valuable insights.

When discussing NOx and its association with mortality, acknowledge the potential limitations of using NOx as a marker for other combustion-related pollutants. Address the need for further research to identify the specific components responsible for health effects.

The section discussing future research and policy implications is well-placed. However, consider a separate section or subsection for strengths and limitations to provide a clear overview of the study's robust aspects and potential sources of bias. The conclusions are well-summarized and reflect the study's findings. However, it could benefit from a more succinct summary of the primary findings to reiterate the key takeaways for the reader. Throughout the text, ensure consistent use of acronyms and abbreviations. For instance, PM10 and PM2.5 should always be spelled out the first time they are used, with the abbreviation in parentheses.

Proofread the manuscript carefully for grammar and spelling errors.

Overall, this scientific study provides valuable insights into the associations between air pollutants and long-term mortality. The manuscript is well-structured and informative, but a few improvements in organization and presentation could enhance its clarity and accessibility to readers.

Author Response

Dear Reviewer!

See the answers in the attached file.

Best regards,

Henrik Olstrup

Reviewer 2 Report

Comments and Suggestions for Authors

What I do not understand is why in Figures 4 and 5 the result of Model PM10+BC and BC+PM10 is not reported in Figure 3. 

The same happens in Tables A2 and A3 where the results of the models  PM10+BC and BC+PM10 are not reported either, but they are reported in Table A1.

These issues were either not explained by the asuthors or I could not find them in the manuascript. Please address that issue.

Author Response

(The authors gave the same response as above.)

Reviewer 3 Report

Comments and Suggestions for Authors

This is an interesting cohort study using a lot of data over many years, which gives a lot power to the study.

I feel the paper is long, and potentially could be shortened, and more concise, without loss of message.

All of the figures are based on "model 3" so does "model 3" need to be included as frequently as it is.

Figure 5 (and in other figures) you have PM2.5 + NOx, and then NOx + PM2.5.  The estimates are quite different.  It is not clearly explained s to why, and what the model is doing to produce such different estimates when using the same two pollutants.  This needs to be checked and clearly explained.

There are good discussion points raised in relation to the lags, and the changes in pollutants over the years.

Have the methods used to measure NOx changed over the years, or have the sites changed.  Proportion of diesel vehicles in the fleet, has this changed much over the study period? Perhaps these could be looked at or commented on.

Author Response

(The authors gave the same response as above.)

Reviewer 4 Report

Comments and Suggestions for Authors

Review comments on “Long-Term Mortality Effects ….”, Olstrup et al., Toxics 2023

Summary

This paper has the potential to make important contributions to the understanding of long-term associations of mortality with ambient quality, chiefly through longitudinal analysis over its 25 years of follow-up. It includes 3 different measures of particulate matter; the exposure modeling approach could have accommodated distinguishing among particles from different source categories. However, the paper lacks clear distinctions between long- and short-term relationships and requires substantial modifications, beginning with trend plots of each pollutant.

Traditionally, “lag” refers to the time required to respond to a (previous) acute exposure, typically a few days. It is not clear how “lag” might pertain to long-term effects. For example, “lag0” refers to deaths during the year of exposure.  Since air quality improved over time, those who survived the longest would have had the lowest exposures. Such results should be compared with previous estimates of short-term (daily) associations, and the BC and NOx results indicate the absence of mortality displacement (harvesting). What does lag1-5 mean? What is a “lag window”?  Does it mean exposures of 5-y duration during the period from 1-5 years before death? Does this also apply to lag6-10? If so, both lag1-5 and 6-10 refer to exposures of 5-y duration, differing only according to the temporal trends in ambient air quality that were not revealed to us. Why should this matter? Was linearity of dose-response considered?

The two main categories of air pollution health studies, short- and long-term, differ in duration of exposure, which is not discussed in this paper. Timing of exposure and hence age of decedent affect exposures. We would like to know how duration of exposure affects risk and whether improvement in air quality during follow-up conferred any benefits. The importance of duration has clearly been shown with respect to smoking (pack-years) and occupation (years of employment); the current study has the potential to extend that concept to ambient air quality. The lag concept is an important part of the study design but its relevance is never discussed. The paper is more concerned with identifying responsible pollutants than with explaining plausible mechanisms of long-term effects that depend on duration.

Other Comments

I would have liked to have seen:

·        IQR values in Table 1.

·        Risk estimates based on duration of exposure.

·        Quantitative comparisons with findings of some of similar papers that were cited.

·        Comparisons of modeled and measured ambient air quality. How do the modeled values of NOx compare with measured values of NO2?

·        Correlations among air pollutants, over time.

·        Discussion of positive effects in combination with negative effects in 2-pollutant models.

This last point requires clarification; the 2-pollutant results must be considered jointly.  For example, in Figure 3, a significant beneficial effect is shown for BC with a significant adverse effect for NOx. The text should explain this; it likely resulted from correlation between BC and NOx. The net effect is the arithmetic sum of the two estimates; they cannot be considered independently. Accordingly, Figure 2 should be presented as 5 different paired estimates with a space between each pair, while it now looks like a group of 10.

On a positive note, I was pleased to see indoor exposures mentioned, which is rare in studies of this type. That there are no indoor sources of traffic-related pollution could also have been mentioned.

Conclusion

This paper cannot be accepted as is, but the dataset appears worthy of further consideration. The authors should be encouraged to abandon their mysterious “lag” concept and focus on time-dependent durations of exposures. The basic question should be: what benefits have accrued during the 25 years of improved ambient air quality following the reductions in emissions? 

Author Response

(The authors gave the same response as above.)

Round 2

Reviewer 4 Report

Comments and Suggestions for Authors

Re-Review Comments on Olstrup et al., Toxics paper 2597533

I am still wholly dissatisfied with the revised treatment of “lag”, which is a major feature of the study design. The rationale for focusing on years 6-10 is not clear nor is the definition of “short” term, which has generally been taken as a few days or weeks. Short-term effects have long been regarded as acute, during which previously ill individuals are pushed over the brink. By contrast, long-term effects are regarded as the development of new cases of disease over decades. Differences among “lags” are shown in the Figures but were not tested for significance, nor are they discussed in the text. These deficiencies must be remedied before publication can be recommended.

1.      What “lag” effects did the authors expect to find?

2.      What did they actually find?

3.      How did those findings differ by pollutant?

4.      What did the authors conclude about causality?

Here are some possibilities:

1.      If effects did not persist more than a year, responses were acute and pertain to previously compromised individuals such as in heat waves.

2.      If effects increased with exposure duration, new cases of diseases developed as with cigarette smoking, poor diet, lack of exercise.

3.      If effects of exposures from years 6-10 exceeded those from years 1-5, responses were non-linear, assuming that exposures decreased over time.

4.      If effects from lags 0 to 10y were not significantly different, responses were independent of exposure duration and thus must have been due to factors other than pollution such as inadequate medical care, indoor exposures, or poor personal habits. Alternatively, the sample may have been too small.

An entirely new section should be added to the paper encompassing the above.

Author Response

Dear Reviewer!

See our answers in the attached file. 

Best regards,

Henrik Olstrup
